# Attenuation of Adverse Postinfarction Left Ventricular Remodeling with Empagliflozin Enhances Mitochondria-Linked Cellular Energetics and Mitochondrial Biogenesis

**DOI:** 10.3390/ijms23010437

**Published:** 2021-12-31

**Authors:** Yang Song, Chengqun Huang, Jon Sin, Juliana de F. Germano, David J. R. Taylor, Reetu Thakur, Roberta A. Gottlieb, Robert M. Mentzer, Allen M. Andres

**Affiliations:** 1Cedars-Sinai Medical Center, Los Angeles, CA 90048, USA; Yang.Song@cshs.org (Y.S.); Chengqun.Huang@cshs.org (C.H.); jjsin@ua.edu (J.S.); Juliana.Germano@cshs.org (J.d.F.G.); davidjrtaylor@outlook.com (D.J.R.T.); Reetu.Thakur@cshs.org (R.T.); Roberta.Gottlieb@cshs.org (R.A.G.); Robert.Mentzer@cshs.org (R.M.M.J.); 2University of Alabama, Birmingham, AL 35294, USA

**Keywords:** empagliflozin, Parkin, mitophagy, mitochondrial biogenesis, adverse remodeling

## Abstract

Sodium–glucose cotransporter 2 (SGLT2) inhibitors such as empagliflozin are known to reduce the risk of hospitalizations related to heart failure irrespective of diabetic state. Meanwhile, adverse cardiac remodeling remains the leading cause of heart failure and death in the USA. Thus, understanding the mechanisms that are responsible for the beneficial effects of SGLT2 inhibitors is of the utmost relevance and importance. Our previous work illustrated a connection between adverse cardiac remodeling and the regulation of mitochondrial turnover and cellular energetics using a short-acting glucagon-like peptide-1 receptor agonist (GLP1Ra). Here, we sought to determine if the mechanism of the SGLT2 inhibitor empagliflozin (EMPA) in ameliorating adverse remodeling was similar and/or to identify what differences exist, if any. To this end, we administered permanent coronary artery ligation to induce adverse remodeling in wild-type and Parkin knockout mice and examined the progression of adverse cardiac remodeling with or without EMPA treatment over time. Like GLP1Ra, we found that EMPA affords a robust attenuation of PCAL-induced adverse remodeling. Interestingly, unlike the GLP1Ra, EMPA does not require Parkin to improve/maintain mitochondria-related cellular energetics and afford its benefits against developing adverse remodeling. These findings suggests that further investigation of EMPA is warranted as a potential path for developing therapy against adverse cardiac remodeling for patients that may have Parkin and/or mitophagy-related deficiencies.

## 1. Introduction

Adverse postinfarction left ventricular (LV) remodeling is a major cause of heart failure (HF) in the United States today [1,2,3]. This is due in part to the welcomed increase in patient survival due to early initiation of reperfusion therapies after acute myocardial infarction (AMI). While angiotensin-converting enzyme inhibitors, angiotensin receptor blockers, beta blockers and sacubitril/valsartan are the pillars of pharmacologic therapy for HF with reduced ejection fraction (HFrEF), the benefits are most often seen in patients with large infarcts and individuals who are not candidates for reperfusion therapies. Their use, however, relates to no more than a 20–25% reduction in major adverse cardiac events (MACE) [4]. Thus, there is clearly a need to develop additional therapeutic strategies.

One approach that holds considerable promise is the use of a relatively new class of drugs used to treat type 2 diabetes mellitus (T2DM), specifically sodium–glucose co-transporter 2 (SGLT-2) inhibitors (glifozin class). A major driver for the use of these drugs in the setting of HF is the EMPA-REG OUTCOME trial [5]. This was the first large randomized clinical trial that showed empagliflozin (EMPA), a specific SGLT-2 inhibitor, was not only effective in reducing MACE but that the outcome was driven primarily by a reduction in cardiovascular deaths and hospitalization for heart failure [5,6,7]. Two additional cardiovascular outcome trials (DAPA-HF (assessing dapagliflozin) and EMPEROR-Reduced (assessing empagliflozin)) demonstrated similar outcomes [8,9,10]. These findings have resulted in recent FDA approval of dapagliflozin and empagliflozin for the treatment of HFrEF. This is despite the fact, however, that the mechanism(s) underlying their effectiveness remains unknown [11]. If SGLT-2 inhibitors are to become a standard of care for heart failure in nondiabetic patients, a much better understanding of their mechanism of action is needed. This is particularly important given the growing evidence that the cardioprotective effect of SGLT-2 inhibitors appears to be mediated independent of glycemic control [12]. Moreover, given that two SLGT2 inhibitors have received FDA approval for therapy and many others are under development it is increasingly important to understand how and why they work—assuming that selectivity among these inhibitors and their adverse side effects will vary [13].

One mechanism that has been recently implicated but not fully explored is the beneficial effect of SGLT2 inhibitors on cardiac mitophagy (autophagy selective for mitochondria), mitochondrial biogenesis and improvements in mitochondrial function [14,15]. To further explore these aspects in a preclinical setting, we assessed the effect of EMPA on postinfarction remodeling in wild-type and mitophagy-compromised Parkin knockout (PKO) mice. Parkin is an E3 ubiquitin ligase found in several tissues including, but not limited to, heart, brain, liver and kidneys [16,17]. A key role of Parkin is in selectively targeting damaged or unwanted mitochondria for elimination through autophagy [18,19,20,21]. The effect of the drug on bioenergetics was also assessed in differentiated H9C2 cardiomyocytes and mitochondria isolated from the hearts of WT and PKO mice. Parkin-mediated mitophagy has been shown to be an important aspect of cardioprotection under various settings [22,23,24,25,26]. An important focus of this work was to evaluate the requirement of Parkin in the mechanism of empagliflozin-mediated attenuation of adverse cardiac remodeling.

## 2. Results

### 2.1. The Effect of EMPA on Postinfarction Adverse LV Remodeling in WT Mice and Immune Cell Infiltration

A schematic of our experimental procedure detailing our mouse model of permanent coronary artery ligation (PCAL) and EMPA treatment scheme is shown in Figure 1A. The timing of when we investigated for key features that occur in MI-induced adverse cardiac remodeling in mouse was inspired by the work of Yang and colleagues [27]. Histological analysis performed at 3 days after PCAL revealed substantially less infiltration of immune cells in the border zone region compared to vehicle control (hydroxyethyl cellulose, HC, Figure 1B). At 14 days post PCAL, treated mice had less fibrosis in the border zone region compared to control vehicle-treated mice (Figure 1C). Correspondingly, vehicle-treated WT mice exhibited greater ventricular wall thinning and were enlarged relative to the hearts of EMPA-treated mice; likewise, the heart weight to body weight and heart weight to tibial length ratios were greater in the control animals (see Figure 1D,E). The EF and FS obtained by echocardiography performed 14 and 28 d post-MI in treated WT mice were significantly better in the EMPA-treated cohort, although improvement did not reach baseline values from naïve mice (EF ~60% and FS ~30%). No differences were observed in calculated LV mass, diastolic and systolic LV volume, LVID and LVPW; however, there was a trend towards improvement in these parameters in the EMPA-treated mice at 28 d (Table 1).

### 2.2. The Effect of EMPA on Mitochondrial Energetics and Content in H9C2 Cardiac Cells

To test the hypothesis that the beneficial effects of EMPA on LV remodeling and cardiac function were due to a direct effect on the cardiomyocytes and improvement in mitochondrial quality, we incubated differentiated H9C2 cells for 24 h with 0.01% *v*/*v* DMSO or EMPA (0.1 or 1 µM). Both doses of EMPA were associated with an improvement in the basal and maximal respiratory capacity of H9C2 cardiomyocytes (Figure 2A–C). Western blot analysis revealed that EMPA was also associated with a marked increase in the surrogate markers for mitochondrial content, such as COX-IV, COX-V subunit A, TOM40 and TOM70 during cotreatment with bafilomycin A1 (to block autophagic flux), suggestive of increased mitochondrial turnover (Figure 2D). The findings also demonstrate that EMPA acts directly on cardiac cells to induce mitochondrial turnover, leading to improved mitochondrial energetics.

### 2.3. Effect of EMPA on Postinfarction Adverse LV Remodeling and Immune Cell Infiltration in Mitophagy-Impaired PKO Mice

A schematic of our experimental procedures detailing our mouse model of PCAL and EMPA treatment in Parkin-knockout mice (PKO) is shown in Figure 3A. Echocardiographic findings obtained 14 days after PCAL are shown in Table 2. EMPA treatment was associated with markedly better EF and FS when compared to their respective vehicle control. Like WT mice, EMPA-treated PKO mice also exhibited less infiltration of immune cells (Figure 3B) in the border zone region compared to vehicle controls. EMPA-treated mice exhibited less fibrosis 14 days after PCAL (Figure 3C).

### 2.4. The Effect of EMPA on Isolated Mitochondria Obtained from WT Mice and PKO Mice with Impaired Mitophagy

Respiratory analysis performed on freshly isolated mitochondria from vehicle- or EMPA-treated WT or PKO mice 14 days post MI allowed for a direct comparison of the respiratory performance between equal amounts of mitochondria between study groups. EMPA increased baseline mitochondrial oxygen consumption and maximal respiratory capacity in WT mice but not PKO mice (Figure 4A,B). Increased respiratory capacity is typically due to an increase in mitochondrial content. Analysis of total heart lysates from WT mice revealed an increase in mitochondrial proteins, including representative subunits of each of the 5 OXPHOS complexes, as well as COX IV, TOM20 and Mfn2 (Figure 5A). A similar increase in mitochondrial protein content was observed in PKO mice as well (Figure 5B), although this was not reflected in a similar increase in mitochondrial respiration (Figure 4C,D). When whole heart lysates were examined for mitochondrial markers as a surrogate for mitochondria content, both WT and PKO mice exhibited a general increase in most of these markers (Figure 5A,B).

## 3. Materials and Methods

### 3.1. Animals

All animal procedures involving C57BL/6J wild-type male mice and Parkin gene knockout male mice (Parkin^−/−^, PKO) were performed in accordance with institutional guidelines and approved by the Institutional Animal Care and Use Committee of Cedars- Sinai Medical Center. The C57BL/6J mice and PKO mice used in these studies were between 12–16 weeks old. At the end of the experimental procedures, mice were anesthetized by isoflurane and euthanized by cervical dislocation.

### 3.2. Animal Protocol for In Vivo Wild-Type and PKO Mouse Studies

Animals were anesthetized with 2% isoflurane gas, intubated and ventilated. Pressure-controlled ventilation (Harvard Apparatus) was maintained at 2–3 cm H_2_O; anesthesia was maintained with 0.5–1% isoflurane throughout the procedure. A permanent coronary artery ligation (PCAL) procedure was performed as previously described [28]. Two hours after surgery, mice were randomized to receive vehicle (0.5% hydroxyethyl cellulose, HC) or 10 mg/kg/day empagliflozin (EMPA) daily via oral gavage for 14 days.

At 14 days post-PCAL, WT and PKO mice underwent echocardiography. Hearts were harvested and weighed to assess wet/dry ratio and heart weights were normalized to tibia length. For biochemistry, the largely acellular fibrous infarct region was excluded. In a separate series of experiments, hearts were harvested 3 and 14 days post-PCAL for histological assessment of immune cell infiltration and fibrosis. A separate cohort of WT mice were followed for an additional 2 weeks (to 28 days post-PCAL) at which time echo was performed and hearts were obtained for the same biochemical studies performed at 14 days. Studies were limited to 14 days post-PCAL in PKO mice due to anticipated high morbidity and mortality rates characteristic of these mice [25,28]. The protocol is summarized in Figure 1A and Figure 3A.

### 3.3. Echocardiography

Transthoracic echocardiography was performed with a high-resolution Vevo 3100 (Fujifilm Visual Sonics, Inc.; Toronto Canada). Mice were anesthetized with an inhaled mixture of oxygen and 2% isoflurane gas at a flow rate of 1 L/min. Heart rate was maintained between 400 to 500 bpm and ultrasound gel was warmed to body temperature before application to the depilated mouse chest. The echocardiography data collection and cardiac function analysis has been validated by others and described in our previous paper [28]. LV volume, ejection fraction (EF)% and Fractional Shortening (FS) were calculated from M-mode-derived LV dimensions using the formula (LV Volume = [7/(2.4 + LVID)] * LVID3 EF = (LV vol,d − LV vol,s)/LV vol,d * 100. Fractional Shortening (FS) = [(LVd − LVs)/LVd)] * 100.

### 3.4. Histology for Immune Cell Infiltration and Fibrosis Post-MI

Three and 14 days after PCAL, hearts were harvested, perfused with PBS, fixed in 4% paraformaldehyde (PFA) for 24 to 48 h and then transferred to 70% ethanol. Tissues were prepared and preserved through cryo-embedding in an optimal cutting temperature compound or paraffin embedding for subsequent histopathology staining and microscopic analysis (Keyence Biorevo BZ-9000). To identify immune cell infiltration on day 3, slide sections were stained with hematoxylin and eosin (Stain Kit Vector Laboratories Inc., H-3502) according to manufacturer’s protocol, followed by dehydration in 95–100% ethanol gradient. Slide sections were covered with hematoxylin and rinsed with distilled water. Adequate Eosin Y solution was applied on sections, incubated for about 3 min and rinsed using 100% ethanol. Slides were dehydrated in 3 changes of 100% ethanol, cleared with xylene and mounted with Cytoseal (VWR, 48212-187).

Fibrosis was assessed at 14 days by staining heart sections with Masson’s Trichrome (Sigma-Aldrich, HT15-1KT). Briefly, slide sections were deparaffinized using xylene and an ethanol gradient (100–70%) and hydrated with deionized water. Sections were placed in a container with Bouin’s solution overnight, washed in running tap water to remove yellow color from sections and stained in Weigert’s iron hematoxylin and Biebrich scarlet-acid fuchsin solution. Further steps included decolorizing with phosphomolybdic -phosphotungstic acid solution, aniline blue solution staining and clarification in 1% acetic acid solution. Sections were dehydrated through alcohol, cleared in xylene and then mounted.

### 3.5. Effect of EMPA on Mitophagy and Mitochondrial Biogenesis on Differentiated H9C2 Cells

H9C2 (rat cardiac myoblast cells) were purchased from ATCC (CRL-1446) and maintained in high-glucose growth media consisting of DMEM (Gibco, 11995-073; Thermo Fisher Scientific, Waltham, MA, USA) supplemented with 10% fetal bovine serum (Life Technologies, 16010-159; Carlsbad, CA, USA) and 1% antibiotic/antimycotic (Life Technologies, 15230-062; Carlsbad, CA, USA). Cells were differentiated for 4 to 5 days in high-glucose DMEM containing 1% fetal bovine serum and 1% antibiotic/antimycotic before starting experiments. To investigate the effect of the SGLT2 inhibitor on autophagy/mitophagy and/or mitochondrial biogenesis, cells were treated with 0.1 or 1 μM EMPA or vehicle 0.1% *v/v* dimethyl sulfoxide (DMSO) (Sigma-Aldrich, D4540; St. Louis, MO, USA) for 24 h. Then, 8 h before cell harvesting, 50 nM Bafilomycin A1 or 0.1% *v*/*v* DMSO was added to inhibit lysosomal degradation for flux determination. Functional respiratory analysis and biochemical analyses of mitochondrial components were then performed.

### 3.6. Subcellular Fractionation of Heart Tissues and Differentiated H9C2 Cells

For subcellular fractionation to obtain whole lysate and mitochondria-enriched heavy membrane, heart tissues were homogenized in 1 mL HES buffer [4-(2hydroxyethyl)-1-piperazineethanesulfonic acid (HEPES) (10 mM, pH 7.4), ethylenediaminetetraacetic acid [EDTA] (1 mM) and sucrose (250 mM)]. Fresh protease inhibitor cocktail (Roche, 05056489001) and phosphatase inhibitors (20 mM sodium fluoride) were added immediately before use. Homogenates were spun down at 1000 g to eliminate nuclei and debris. Next, 200 μL of postnuclear supernatants were saved as whole lysate; the remaining supernatants were then spun down at 7000 g to obtain the mitochondria-enriched heavy membrane pellet. The pellet was then resuspended in RIPA buffer pH 8.0 [Trizma base (50 mM; Sigma-Aldrich, T1503), NaCl (150 mM; Sigma-Aldrich, S7653), ethylene glycol tetraacetic acid [EGTA] (2 mM; Sigma-Aldrich, E4378), ethylenediaminetetraacetic acid [EDTA] (1 mM; Sigma-Aldrich, E4884), sodium deoxycholate (0.5%; Sigma-Aldrich, D6750), NP-40 (1%; Sigma-Aldrich, I3021), sodium dodecyl sulfate [SDS] (0.1%; Bio-Rad Laboratories Inc., 161–0302)] with protease and phosphatase inhibitors added just before use.

For the in vitro H9C2 cell experiments, media were removed, and cells were rinsed with PBS. HES buffer with protease and phosphatase inhibitors was then added and cells were collected with a cell scraper and transferred to an Eppendorf tube. Mitochondria fractionation from differentiated H9C2 cells followed the protocol previously described [28]. All the cell samples and tissue samples were stored at −80 °C until use.

### 3.7. Western Blot Analysis

Protein concentrations of whole lysates, mitochondria-enriched heavy membrane and nuclear samples were determined by Bio-Red DC^TM^ Protein Assay kit (BIO-RAD, 5000111; Hercules, CA, USA). Equal amounts of proteins were separated on Bolt 4–12% Bis-Tris Plus gels (Thermo Fisher Scientific; Waltham, MA, USA) and transferred to nitrocellulose membranes. Membranes were blocked with 5% nonfat dry milk prepared in Tris-buffered saline with 0.1% Tween 20 (TBS-T) pH 7.6 for 1 h and incubated with 1:1000 diluted primary antibodies against: P62, TOM70, TOM40, COX IV, Mfn2, PINK1, PGC-1α and PPAR-α. After primary incubation, blots were washed with TBS-T at room temperature and incubated with KPL Peroxidase Labeled Goat anti-Mouse IgG (H + L) (1:3000, KPL Affinity Purified Antibody, SeraCare) or KPL Peroxidase-Labeled Goat anti-Rabbit IgG (H + L) secondary antibodies for 2 h in room temperature. Membranes were washed in TBS-T (3 × 5 min), developed with Clarity Western ECL Substrate (Bio-Rad) and imaged using a ChemiDoc XRS and Image Lab software v 5.0 (Bio-Rad). Densitometry was performed using NIH Image J software v 1.51.

### 3.8. Seahorse Mitochondria Stress Test (Respirometry)

A mitochondria stress test was carried out on differentiated H9C2 cells using XFe24 extracellular flux analyzers. First, 40,000 cells were seeded in each well of Seahorse XFe24 cell culture microplate and began differentiation on the second day. Cells were differentiated for 5 days and pretreated with 0.1% DMSO or 0.1 µM/1 µM Empagliflozin for 24 h. Seahorse assay media was freshly prepared with Seahorse XF base media with 1 mM sodium pyruvate, 10 mM glucose and 2 mM glutamine at pH7.4. The cell culture media was replaced with Seahorse assay media and the cell plate was stored in 37 °C incubator (no CO_2_) for 45–60 min before the assay. Mitochondrial respiration was measured following sequential injection of 1 µM oligomycin, 1 µM FCCP and 0.5 µM/0.5 µM antimycin/rotenone.

An isolated mitochondria stress test was conducted using a Seahorse XFe96 extracellular flux analyzer. Fourteen days after PCAL, hearts were harvested from C57BL/6J wild-type and PKO mice and homogenized with Polytron in HES buffer with 0.2% fatty-acid-free bovine serum albumin (BSA). Samples were spun down at 1000× *g* for 5 min to eliminate nuclei and debris; supernatants were further spun down at 7000× *g* for 10 min to obtain mitochondria-enriched heavy membranes. Mitochondria pellets were resuspended in HES buffer. Protein concentration of samples were measured using Bio-Rad DC^TM^ protein assay kit. Next, 3 μg isolated mitochondria in 25 μL HES buffer were loaded into XFe96 cell culture plates and spun down at 2000× *g* for 20 min, then 155 μL of mitochondrial assay solution (MAS) with 0.2% fatty acid-free BSA was added to bring the volume up to 180 μL. MAS buffer consists of 70 mM sucrose, 220 mM mannitol, 5 mM KH_2_PO_4_, 5 mM MgCl_2_, 2 mM HEPES, 1 mM EGTA and 0.2% essentially fatty-acid-free BSA, pH 7.2. The plate was warmed to 37 °C for 10 min prior to the assay in incubator (no CO_2_). Oxygen consumption was subsequently monitored following sequential injection of 5 mM pyruvate, 5 mM malate and 0.25 mM adenosine diphosphate; 1 µM oligomycin; 1.5 µM FCCP and 0.5 µM/0.5 µM antimycin/rotenone. Rates of mitochondrial oxygen consumption were normalized to protein loaded.

### 3.9. Statistical Analysis

A standard student’s *t*-test was used to determine statistical significance with *p*-values less than 0.05 accepted as significant. Error bars indicate the standard deviation. All values are presented as mean ± standard deviation. All the statistical analysis was performed on GraphPad Prism v.6.

## 4. Conclusions

The efficacy of the sodium-glucose transporter 2 (SGLT-2) class of inhibitors in reducing MACE in diabetic patients is strongly supported by the findings of several randomized large-scale clinical trials (EMPA-REG OUTCOME, CANVAS, DAPA-HF, DECLARE-TIMI, VERTIS-CV) [5,9,29,30,31]. Clearly, the mechanism underlying this salutary effect goes beyond enhanced glucose control [12,32]. This is underscored by the EMPEROR-Reduced trial wherein empagliflozin was found to be superior to placebo for preventing CVD-associated death and hospitalizations for worsening heart failure, irrespective of presence or absence of diabetes [33]. While several mechanisms have been proposed, including enhanced natriuresis, better blood-pressure control, favorable changes in the renin–angiotensin–aldosterone axis, improved renal function and less oxidative stress [10,11], there is little, if any, evidence that any one of these or combination of these explain the magnitude of efficacy of this class of drugs in mitigating the progression of heart failure. The salient findings in this study and our earlier work offer some concrete clues, however.

Previously, we characterized the utility of another compound from a different class of drugs shown to be effective in treating chronic progressive heart failure in diabetic patients. The compound is a unique small molecule GLP-1-like receptor (GLP1R) agonist known as Compound C or 2-quinoxalinamine, 6,7-dichloro-N-(1,1-dimethylethyl)-3-(methylsulfonyl)-6,7-dichloro-2-methylsulfonyl-3-N-tert-butylaminoquinoxaline (DMB) [20]. We found that it was very effective in mitigating infarction-induced adverse remodeling via its stimulatory effect on mitophagy and mitochondrial turnover, but the benefit was lost in PKO mice [28]. Given the reports that GLP1R agonists may have similar cardioprotective properties to the SGLT2 inhibitors [34], we hypothesized that a mechanism underlying the EMPA effect would be similar to the GLP1R agonist. However, in our current report we demonstrated that while enhancing mitochondrial energetics may be common to both drugs, EMPA is unique in that it appears to not require Parkin, and by extension a robust mitophagy capacity to elicit its beneficial effects against adverse remodeling.

In our animal studies here, EMPA was effective in mitigating adverse postinfarction LV remodeling when administered post-PCAL (permanent coronary arterial ligation) challenge. PCAL was associated with adverse remodeling and deterioration in cardiac function in both WT and PKO mice, which was attenuated with EMPA administration. The beneficial effects of EMPA were sustained over the course of 28 d in the WT mice despite discontinuation of the drug 14 d prior to the last assessment of cardiac function. The salutary effects paralleled a decline in the degree of the acute immune cell infiltration assessed 3 d after MI and was associated with a decrease in the magnitude of fibrosis at 14 d in both WT and PKO mice. Unlike our studies using DMB (GLP1R agonist) earlier, in this study EMPA was effective in PKO mice and its benefits appeared to be linked to an increase in mitochondrial biomass without an improvement in mitochondrial quality. Additionally, in differentiated H9C2 cardiomyocytes, EMPA treatment resulted in an increase in basal and maximal respiratory capacity and an increase in the markers of mitochondrial biogenesis. The effects in H9C2 cells were paralleled in vivo and suggest that EMPA has direct effects at the cellular level of the cardiomyocyte. These findings support the contention that EMPA increases both mitochondrial mass and function. Correspondingly, Mizuno and colleagues previously demonstrated that EMPA can normalize the size and number of mitochondria in the context of an ischemic challenge such as I/R which normally causes mitochondrial fragmentation and turnover [35]. In renal cell models, others have found that prevention of mitochondrial fragmentation is a feature of EMPA administration, likely through repression of DRP1 through AMPK activation [36,37,38].

In summary, the beneficial effects of EMPA on adverse postinfarction remodeling may be mediated in part by the drug having a direct effect on mitochondrial quantity in both WT and PKO mice, and additionally mitochondrial quality in WT mice only. The novelty of our work demonstrates that the cardiovascular benefit of empagliflozin in attenuating ischemia-mediated adverse cardiac remodeling is that the mechanism is not fully dependent on Parkin and can be realized without improving the quality of individual mitochondria. Recent work by others investigating the beneficial effects of EMPA in various cardiovascular settings all align with our work in demonstrating that this SGLT-2 inhibitor improves mitochondrial energetics [39,40,41,42]. An important area to focus attention for future studies not examined here would be to investigate the role of known mitochondrial regulators such as PGC-1α and PPARα. Another interesting area of investigation beyond our work here is to elucidate the potential link between mitochondrial energetics, cardiac substrate utilization and the development of MI-induced adverse LV remodeling.

Our findings highlight the efficacy of EMPA in mitigating adverse postinfarction LV remodeling and show that this can occur even in the setting of a diminished capacity to turn over mitochondria. Should this turn out to be a defining mechanism, it is possible that the efficacy of other or newer SLGT2 inhibitors will be determined by their ability to enhance mitochondria-linked cellular energetics. Although controversial depending on the study setting, most agree that MI-induced failing heart leads to a decrease in fatty acid use with a corresponding increase in glucose utilization. This suggestion is supported by the growing list of others who have shown that gliflozin class drugs help to maintain the heart’s preference for utilizing fatty acids and ketone bodies [15,43,44,45]. However, in most cases heart failure is associated with a decrease in overall cardiac energetics [46,47]. These metabolic changes may be directly linked to the degree and direction the heart takes in developing adverse remodeling leading to heart failure. In consideration of our findings here, and the literature surrounding the mechanism of the gliflozin class of drugs in attenuating MI-induced adverse remodeling, we hypothesize that a key feature that underlies their benefits lies in their ability to stimulate mitochondrial-linked energetics (Figure 6). Our findings, along with others, help establish that gliflozins are highly effective at attenuating the onset of adverse cardiac remodeling and development of heart failure beyond their role as anti-diabetic drugs.

## Figures and Tables

**Figure 1 ijms-23-00437-f001:**
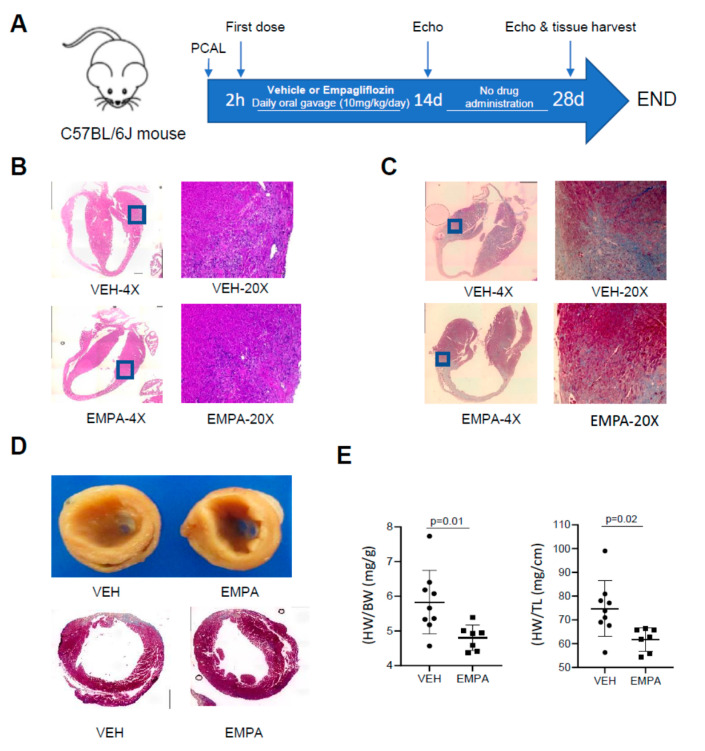
EMPA limits infarct-mediated development of adverse remodeling in WT mice. Age-matched WT mice underwent PCAL and 2 h later were given vehicle (0.5% *w/w* hydroxyethyl cellulose) or EMPA (10 mg/kg/day) via oral gavage daily. (**A**) Schematic of the protocol; (**B**) representative 4× and 20× magnification of heart sections 3d after PCAL, stained with H&E; 20× enlargement of the border zone to highlight infiltration of immune cells; (**C**) representative 4× and 20× magnification of heart sections with Masson Trichrome staining 14 days after PCAL showing fibrosis; (**D**) representative 20× magnification of heart sections with Masson Trichrome staining 28 days after PCAL showing fibrosis and LV dilation; (**E**) heart weight/body weight (mg/g); heart weight/tibia length (mg/cm), *n* ≥ 8.

**Figure 2 ijms-23-00437-f002:**
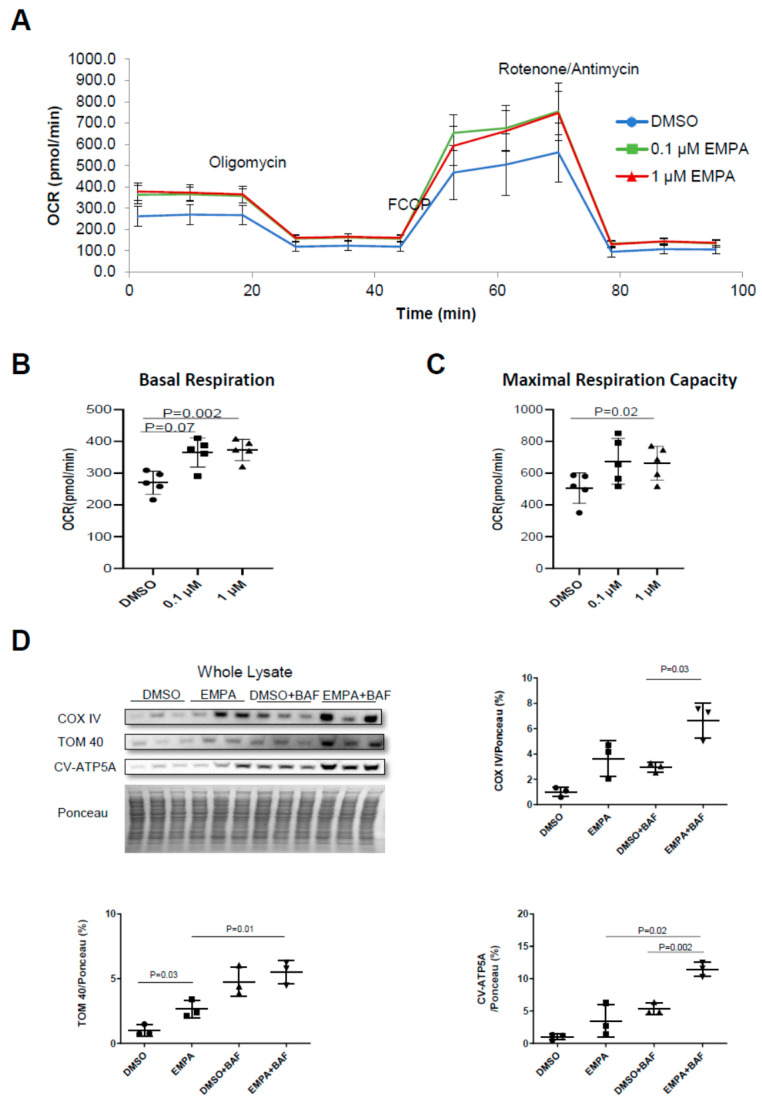
EMPA directly improves mitochondrial quality in H9C2 cells. H9C2 cells were pretreated with DMSO (0.01% *v*/*v*), 0.1 μM or 1 μM EMPA for 24 h, then oxygen consumption rates were measured with seahorse mito stress test. (**A**) Mito stress test respirometry profile of DMSO or EMPA-treated cells; (**B**) quantification of basal respiration; (**C**) quantification of maximal respiration capacity (*n* = 3). (**D**) Western blot analysis and quantification of mitochondria biogenesis markers COX IV, CV-ATP5A and TOM40 in whole cell lysates (*n* = 3).

**Figure 3 ijms-23-00437-f003:**
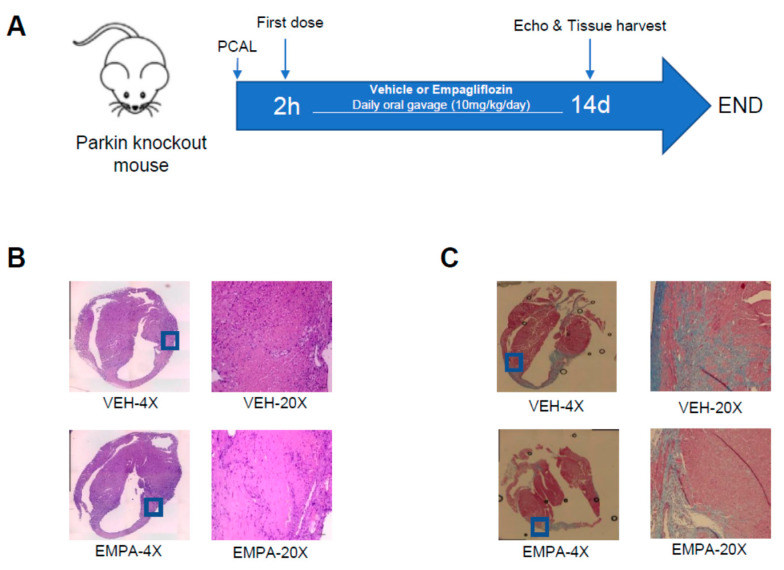
EMPA limits infarct-mediated development of adverse remodeling in Parkin knockout mice. Age-matched WT mice underwent PCAL and 2 h later were given vehicle (0.5% *w/v* hydroxyethyl cellulose) or EMPA (10 mg/kg/day) via oral gavage daily. (**A**) Schematic of the protocol; (**B**) representative 4× and 20× magnification of heart sections stained with H&E of infarct border zone from 3 days after PCAL to show infiltration of immune cells; (**C**) representative 4× and 20× magnification of heart sections with Masson Trichrome staining 14 days after PCAL showing fibrosis.

**Figure 4 ijms-23-00437-f004:**
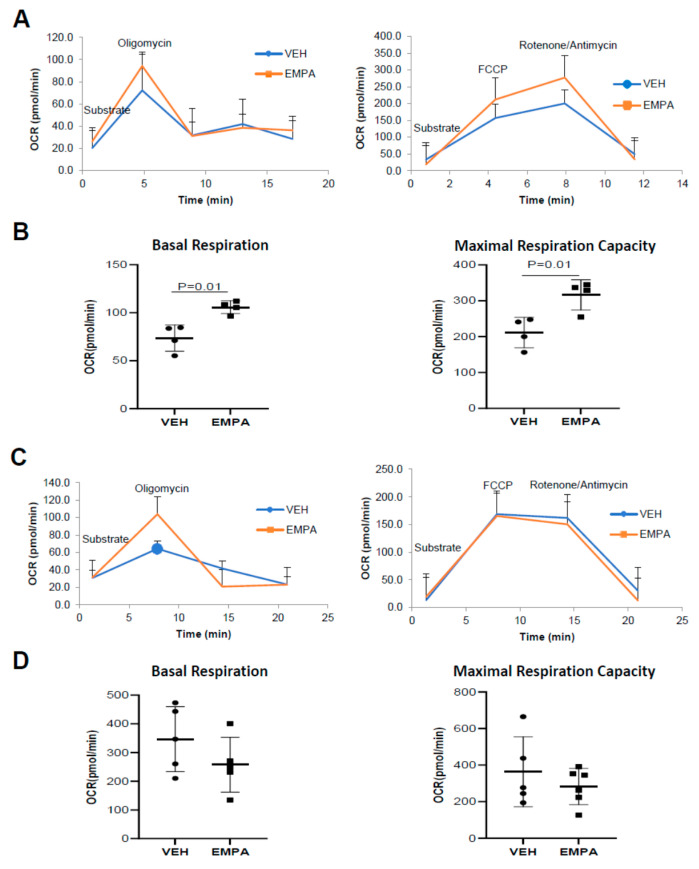
Post-infarction treatment with EMPA boosts mitochondria-linked cardiac energetics of WT but not PKO mice. Mice were subjected to PCAL and treated with vehicle (0.5% *w/w* hydroxyethyl cellulose VEH) or EMPA (10 mg/kg/day). Mitochondria isolated from heart tissue 14 days after PCAL and oxygen consumption was measured using Seahorse mito stress test. (**A**) Respiration profile of isolated mitochondria from WT mice treated with VEH or EMPA; (**B**) quantification of basal respiration and maximal respiratory capacity of mitochondria from WT mice (*n* = 4); (**C**) respiration profile of isolated mitochondria from PKO mice treated with VEH or EMPA; (**D**) quantification of basal respiration and maximal respiration capacity of mitochondria from PKO mice (*n ≥* 5).

**Figure 5 ijms-23-00437-f005:**
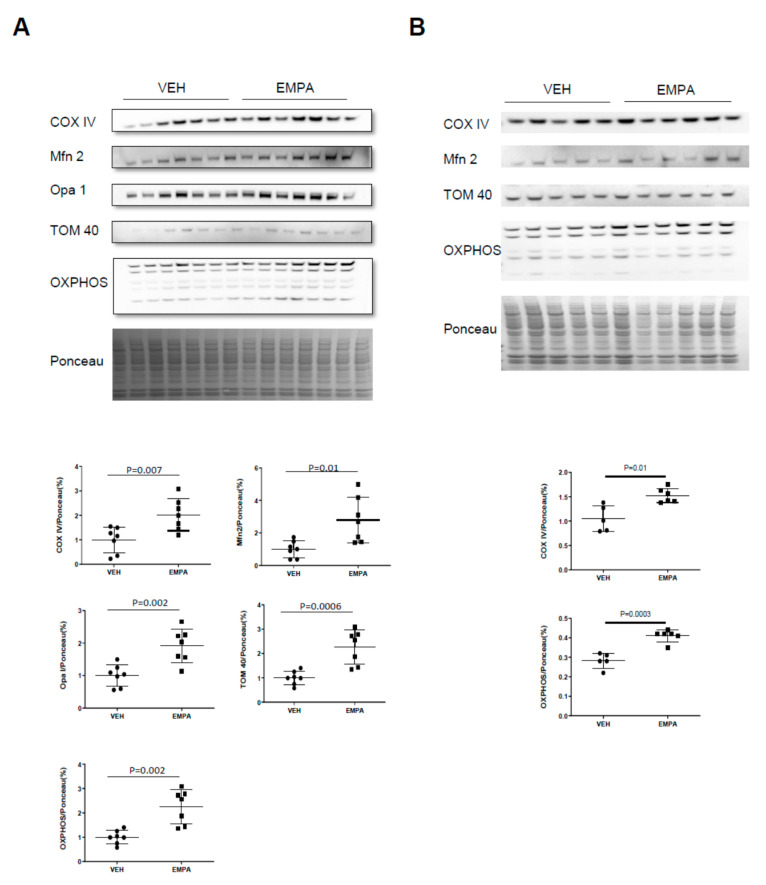
EMPA increases cardiac mitochondrial content in vivo. Mice were subjected to PCAL and then treated with vehicle (0.5% *w*/*v* hydroxyethyl cellulose) or EMPA (10 mg/kg/day) for 14 d followed by tissue harvest. (**A**) Cardiac lysates from WT mice were analyzed by Western blot analysis and quantification of mitochondrial markers COX IV, Mfn2, Opa 1, TOM 40 and OXPHOS (*n* = 7); (**B**) cardiac lysates from PKO mice were analyzed by Western blot analysis and quantification of mitochondrial markers COX IV, Mfn2, TOM 40 and OXPHOS (*n* ≥ 5).

**Figure 6 ijms-23-00437-f006:**
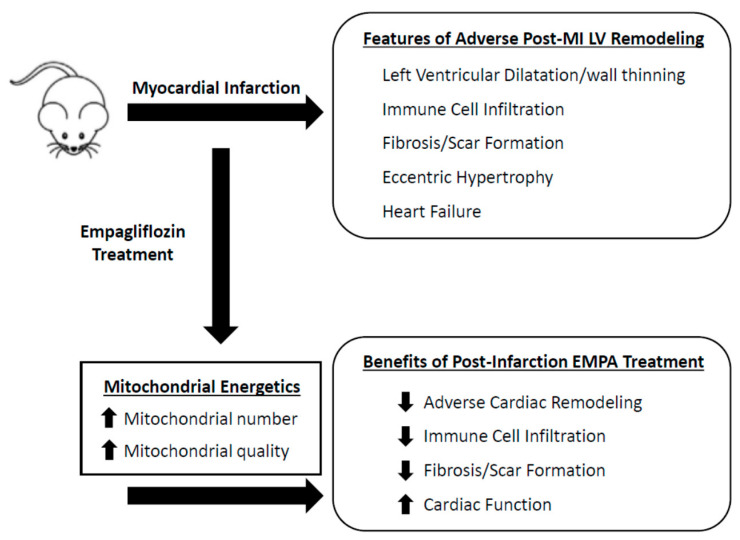
Empagliflozin attenuates infarct-induced adverse cardiac remodeling. Post-MI administration of empagliflozin leads to improved mitochondrial energetics by increasing mitochondrial biogenesis and enhancing the quality of the mitochondrial population in cardiac cells. This is associated with decreasing immune cell infiltration to the border zone, decreasing fibrosis and scar formation and ultimately attenuating adverse remodeling of the heart and delaying the onset of heart failure.

**Table 1 ijms-23-00437-t001:** Assessment of cardiac function and post–infarction cardiac remodeling by echocardiography 14 and 28 days after MI in wild type mice treated with EMPA (10mg/kg/day) or vehicle.

	14 Day			28 Day		
	WT-VEH(7)	WT-EMPA(8)	*p* Value	WT-VEH(6)	WT-EMPA(7)	*p* Value
HR (bpm)	451 ± 70.48	429 ± 49.65	0.49	440 ± 71.57	397 ± 35.10	0.19
EF (%)	38.71 ± 5.76	50.42 ± 8.34	0.01	34.40 ± 8.32	47.85 ± 11.16	0.03
FS (%)	18.70 ± 3.02	25.74 ± 5.21	0.01	16.51 ± 4.37	24.26 ± 6.53	0.03
LV Mass (mg)	152.52 ± 21.22	156.42 ± 57.44	0.87	136.15 ± 34.31	127.01 ± 32.33	0.63
LV Mass (Corrected) (mg)	122.02 ± 16.98	125.13 ± 45.95	0.87	136.15 ± 34.31	127.01 ± 32.33	0.63
LV Vol; d (μL)	93.87 ± 14.75	97.00 ± 25.34	0.78	119.96 ± 61.41	100.99 ± 36.56	0.51
LV Vol; s (μL)	58.18 ± 14.61	48.65 ± 17.09	0.27	81.43 ± 51.59	54.54 ± 26.76	0.25
LVID; d (mm)	4.52 ± 0.30	4.57 ± 0.48	0.83	4.93 ± 0.97	4.61 ± 0.72	0.51
LVID; s (mm)	3.68 ± 0.38	3.40 ± 0.48	0.23	4.14 ± 0.99	3.52 ± 0.75	0.22
LVPW; d (mm)	0.92 ± 0.10	0.88 ± 0.13	0.54	0.86 ± 0.20	0.95 ± 0.18	0.41
LVPW; s (mm)	1.01 ± 0.12	1.15 ± 0.14	0.05	1.01 ± 0.21	1.20 ± 0.17	0.10

**Table 2 ijms-23-00437-t002:** Assessment of cardiac function and post–infarction cardiac remodeling by echocardiography 14 after MI in PKO mice treated with EMPA (10mg/kg/day) or vehicle.

	PKO-VEH(7)	PKO-EMPA(9)	*p* Value
HR (bpm)	410 ± 35.29	385 ± 12.65	0.24
EF (%)	36.28 ± 7.75	46.58 ± 10.78	0.04
FS (%)	17.70 ± 7.75	23.44 ± 6.34	0.05
LV Mass (mg)	144.52 ± 39.27	145.2 ± 42.29	0.60
LV Mass (Corrected) (mg)	115.62 ± 31.42	116.16 ± 33.83	0.60
LV Vol; d (μL)	98.53 ± 35.39	99.04 ± 39.54	0.60
LV Vol; s (μL)	64.84 ± 27.38	56.23 ± 33.19	0.29
LVID; d (mm)	4.56 ± 0.74	4.57 ± 0.75	0.56
LVID; s (mm)	3.78 ± 0.80	3.54 ± 0.86	0.26
LVPW; d (mm)	0.83 ± 0.27	0.79 ± 0.17	0.63
LVPW; s (mm)	1.04 ± 0.27	1.08 ± 0.26	0.77

## Data Availability

Not applicable.

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
