# Peer review of "Attenuation of Adverse Postinfarction Left Ventricular Remodeling with Empagliflozin Enhances Mitochondria-Linked Cellular Energetics and Mitochondrial Biogenesis"

_ijms, 2021, doi:10.3390/ijms23010437_

Round 1

Reviewer 1 Report

The authors have described the empagliflozin effect on ventricle remodeling through mitochondrial biogenesis in a mice model with or without parkin knock out., 28 days after coronary artery ligation. Moreover, some methodology was performed in H9C2 (rat cardiac myoblast cells). Although there are several studies regarding empagliflozin mechanisms on animal models for demonstrating its effects on mitochondrial biogenesis and energy efficiency (Yurista SR, Silljé HHW, Oberdorf-Maass SU, Schouten EM, Pavez Giani MG, Hillebrands JL, van Goor H, van Veldhuisen DJ, de Boer RA, Westenbrink BD. Sodium-glucose co-transporter 2 inhibition with empagliflozin improves cardiac function in non-diabetic rats with left ventricular dysfunction after myocardial infarction. Eur J Heart Fail. 2019 Jul;21(7):862-873. ), this is the first time that these analysis were performed in parkin knock out models. This point is the novelty of the manuscript. 

The manuscript is well written. The authors have used adequate methodology for getting the main results. However, there are several important issues that the authors should improve:

  1. The mitochondrial width and area in myocardium from mice with and without empagliflozin might improve the manuscript.
  2. The inflammatory staining in myocardium from mice with and without empa treatment.
  3. The results regarding (PGC-1α, PPAR-α) were not shown. The authors should demonstrate differences regarding these two proteins before and after empagliflozin treatment.
  4. Is possible to represent the mitochondrial genesis regarding apoptotic markers?

Some graphs are represented by bars and others by dot plots. Individual values with dot plots are more visual.

The authors should represent empagliflozin mechanism after myocardial infarction with respect to their results.

Minor comments: figure word is wrong into text 

Reviewer 2 Report

Thank you for attending to all my comments.

Reviewer 3 Report

Reviewer comments and suggestions

The current manuscript provide to understand the mechanisms which is responsible for the beneficial effects of SGLT2 inhibitors against heart attack.

In this study the authors determine if the mechanism of the SGLT2 inhibitor empagliflozin (EMPA) in ameliorating adverse remodeling was similar and/or to identify what differences exists, to their previous finding.

The study carry out permanent coronary artery ligation to induce adverse remodeling in wild-type and Parkin knockout mice and examined the progression of adverse cardiac remodeling with or without EMPA treatment over time. Similar to GLP1Ra, they observed EMPA affords a robust attenuation of PCAL-induced adverse remodeling. The finding suggested to be different protein and mechanism.

Decision: Major comments

Below are the comments for this paper to be incorporated in the revised version of the manuscript. 

  1. Please change the title of the MS who would be a little friendly to the common reader of your paper, after going through the introduction, I realized the author mentioned specifically SGLT-2 in his study and compare it with the previously published study.
  2. The last para needed to mention that why the study needed to be done. It looks general. It is difficult for me to get though the introduction, try to make it clear with your proposed study
  3. Some provided figures are blurred
  4. Discussion should be mentioned in the manuscript
  5. Line 317 what the authors want to mention here
  6. It is better that in the first para of discussion, the author needs to highlight the novel finding of the study
  7. Please check the journal guidelines of MDPI, the provided all references need to be modified.

Round 2

Reviewer 1 Report

The authors have improved the manuscript. However, the authors did not perform electron microscopy for testing mitochondria shape nor inflammatory cells infiltration. The authors have to make a figure drawing the associated mechanism of empagliflozin according their results. 

Reviewer 3 Report

No more comments

Author Response

We thank the reviewer once again for your comments previously which has helped us to improve our manuscript.